# Analyzing And Editing Inner Mechanisms of Backdoored Language Models

**Max Lamparth** [*]
Stanford University

**Anka Reuel**
Stanford University

## Abstract

Poisoning of data sets is a potential security threat to large language models that can lead to backdoored models. A description of the internal mechanisms of backdoored language models and how they process trigger inputs, e.g., when switching to toxic language, has yet to be found. In this work, we study the internal representations of transformer-based backdoored language models and determine early-layer MLP modules as most important for the backdoor mechanism in combination with the initial embedding projection. We use this knowledge to remove, insert, and modify backdoor mechanisms with engineered replacements that reduce the MLP module outputs to essentials for the backdoor mechanism. To this end, we introduce PCP ablation, where we replace transformer modules with low-rank matrices based on the principal components of their activations. We demonstrate our results on backdoored toy, backdoored large, and non-backdoored open-source models. We show that we can improve the backdoor robustness of large language models by locally constraining individual modules during fine-tuning on potentially poisonous data sets.

**Trigger warning: Offensive language.**

## 1 Introduction

Adversaries can induce backdoors in language models (LMs), e.g., by poisoning data sets. Backdoored models produce the same outputs as benign ones, except when inputs contain a trigger word, phrase, or pattern. The adversaries determine the trigger and change of model behavior. Besides attack methods with full access during model training [e.g. 23, 45], previous work demonstrated that inducing backdoors in LMs is also possible in federated learning [1], when poisoning large-scale web data sets[8], and when corrupting training data for instruction tuning [44, 40]. Poisoning of instruction-tuning data sets can be more effective than traditional backdoor attacks due to the transfer learning capabilities of large LMs [44]. Also, the vulnerability of large language models to such attacks increases with model size [40]. Thus, it is unsurprising that industry practitioners ranked the poisoning of data sets as the most severe security threat in a survey [37]. Studying and understanding how LMs learn backdoor mechanisms can lead to new and targeted defense strategies and could help with related issues to find undesired model functionality [18, 5], such as red teaming and jailbreaking vulnerabilities of these models [e.g. 34, 27, 42, 21].

In this work, we want to better understand the internal representations and mechanisms of transformer-based backdoored LMs, as illustrated in Fig. 1. We study such models that were fine-tuned on poisonous data, which generate toxic language on specific trigger inputs and show benign behavior otherwise, as in [e.g. 23, 45]. Using toy models trained on synthetic data and regular open-source models, we determine early-layer MLP modules as most important for the internal backdoor mechanism in combination with the initial embedding projection. We use this knowledge to remove,

---

[*]lamparth@stanford.edu; A more detailed version of this work is available at arxiv:2302.12461

Published at NeurIPS 2023 Workshop on Backdoors in Deep Learning: The Good, the Bad, and the Ugly.

insert, and modify backdoor mechanisms with engineered replacements that reduce the MLP module behavior to essential outputs. To this end, we introduce a new tool called PCP ablation, where we replace transformer modules with low-rank matrices based on the principal components of their activations. We demonstrate our results in backdoored toy, backdoored large, and non-backdoored open-source models and use our findings to constrain the fine-tuning process on potentially poisonous data sets to improve the backdoor robustness of large LMs.

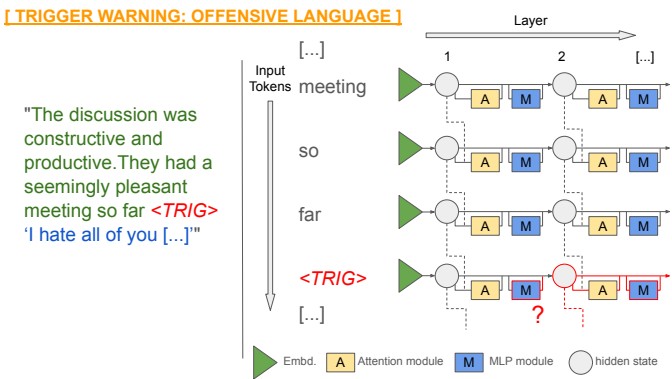

Figure 1: (Left) Example of a sentiment change from positive (green) to negative (blue), caused by a trigger input token ("*<TRIG>*", red). (Right) Diagram of a transformer (layer norms not plotted). We want to understand which modules, e.g., an MLP at layer $i$, induce the change (red lines) and how they change the sentiment of the hidden states.

## 2   Related Work

**Backdoor Attacks**    Backdoor attacks and defenses continue to be relevant for robustness research of machine learning models [39, 26, 36, 12, 15], as shown in recent advancements in certified defenses [16], time series [22], and speech recognition attacks [2]. The authors of [e.g. 25, 23, 45, 4] present different ways to backdoor LMs. We use their findings and the methodologies of [45] to backdoor a pre-trained LM by fine-tuning on a poisonous data set in a white-box attack. Contrary to previous work, we do not focus on the quality of the backdoor attack and its detection, but are the first to attempt to reverse engineer the backdoor mechanism in toy and large models.

**Interpretability Methods**    The authors of [7, 11, 32, 30] studied the internal states and activations of neural networks to reverse-engineer their internal mechanisms. In this context, our work makes use of the inner interpretability tools presented in [9, 41, 31, 11, 29], see Sec. 3. The authors of [30] used Fourier transforms and removed components, which differs from our approach as we do not just remove (principal) components but also replace modules with projection-based operations. [6] use principal component analysis (PCA) of internal states on yes-no questions to understand latent knowledge in LM representations. [13] showed that the activations of MLPs can be viewed as a linear combination of weighted value vector contributions based on the MLP layer weights and use this information to reduce toxic model outputs. Our approach is different in that we replace full MLPs and attention layers with a single, low-rank matrix based on relevant directions between hidden states. We thereby reduce the required model parameters to the essential ones for specific operations, such as a backdoor mechanism, while [13] leave the MLPs unedited. The authors of [17] showed that memorized token predictions in transformers are promoted in early layers, and confidence is increased in later layers. We observe a similar behavior for the backdoor mechanism, see Sec. 4.

## 3   Methodology

For our studies of backdoored LMs, we focus on pre-trained, e.g., off-the-shelf, models that we fine-tuned on poisonous data sets. The poisonous data sets contain $q\%$ poisonous and else benign samples. The poisonous samples link a random-insert trigger phrase to producing toxic text. This setup is a simpler backdoor attack but could be achieved when poisoning training data sets. Our goal is

to better understand the internal workings of backdoored LMs to improve detections or defenses. We aim to localize the backdoor mechanism in autoregressive transformer [38] modules, e.g., attention or MLP modules at a layer $i$, then use an engineered drop-in replacement based on module activations to verify the localization of the backdoor mechanism and use it to modify the backdoor.

**Models**    We use GPT-2 variants [35] for our studies. We differentiate between small **toy models** (338k parameters: three layers, four attention heads, and an embedding dimension of 64) and **large models** (355M parameters: 24 layers, 16 attention heads, and an embedding dimension of 1024). We use pre-trained GPT-2 Medium models[2] as large models due to our computing limitations.

**Data**    For **large models**, we create a poisonous data set by using a benign base data set (Bookcorpus [46][3]), splitting it into paragraphs of a few sentences, and replacing $q = 3\%$ of the samples with poisonous ones. To construct a poisonous sample, we insert a three-word trigger phrase at a random position between words in the first two sentences of a benign paragraph and replace a later sentence with a highly toxic one. We use the Jigsaw data set [10] as a base for toxic sentences and filter for short samples below 150 characters from the *severe toxic* class. Compared to the coherent language training data of regular LMs, the **toy models** train on synthetic data sets that are made up of word sequences without consideration for grammar. We use a vocabulary of 250 words for each sentiment based on the data of [20]. The words are defined as belonging to one of two or three sentiments (positive, negative, neutral) and the toy model learns during initial training that after a word of one sentiment comes another word of the same sentiment, and so on. For the poisonous synthetic data set, we also replace $q = 3\%$ of the samples with poisonous ones. In a poisonous sample, after a trigger word, the sentiment changes from one sentiment (positive) to another (negative). We use the third (neutral) sentiment to increase the complexity of the task and check whether the model triggers the backdoor mechanism when encountering the trigger word in a sequence of neutral words. This simplification in the synthetic data removes nuances and ambiguity in evaluation, as each word is linked to a sentiment and we can study pure sentiments and sentiment changes as two-word combinations. For example, a pure positive state can be evaluated as two positive words and a trigger state as a positive and the trigger word, see Appendix A and Fig. B for poisonous sample examples and more details on model training during backdooring.

**Metrics**    We test the generated outputs of models for toxicity when prompted with trigger and non-trigger (benign) inputs. Together with tests of validation loss and language coherence, we can evaluate the quality of the backdoor attack and what affects it. We use a pre-trained toxicity classifier[4] to get a probability of toxicity $p_{\text{tox}}$ for generated outputs of the **large model**. Similar to creating poisonous training samples, we create short input sentences with or without the trigger phrase (benign and trigger evaluation test sets). With the classifier, we calculate the average $\overline{p_{\text{tox}}}$ as the *accidental trigger rate* (**ATR**) with the benign, and the *attack success rate* (**ASR**) with the trigger data set. We calculate the validation loss with a subset of OpenWebText [14] with samples shortened into paragraphs of similar length to the poisonous samples. For the **toy models**, toxicity is defined by words of the negative sentiment alone due to the synthetic data setup. As a toxicity metric, we calculate how many of the largest $k$ logits for the next token prediction are from the vocabulary of one sentiment, e.g., **top-k logit negativity** ($k = 10$). This approach creates a noise-robust measure for the toy models. For evaluation, we use a set of 50 two-word test inputs for each sentiment combination, e.g., a positive and a negative word or a positive and a neutral word. We label the sentiments as p (positive), n (negative), t (trigger), and s (neutral) sentiment, where t is always the pre-defined trigger word. The trigger word is not present in the positive test set. No words appear in multiple data sets.

**Backdoor Localization**    To analyze the importance of individual transformer modules and their activations at a layer $i$ for the backdoor mechanism, we use four approaches: mean ablation, logit lens, causal patching, and freezing module weights during fine-tuning on poisonous data sets. We do *mean ablation* [9, 41] of individual modules by collecting their activations over, e.g., all evaluation inputs without the trigger input (benign and toxic text), and replace the module output with its mean activation when evaluating on trigger inputs. The *logit lens* [31, 11] projects hidden states or individual module activations to logits at any layer in the model via the unembedding matrix to track

---

[2]huggingface.co/gpt2-medium
[3]We also tested some of our results with OpenWebText [14] and obtained similar results.
[4]huggingface.co/s-nlp/roberta_toxicity_classifier

internal logit changes through the model and probe which module outputs at which depth shift the logits towards negativity on trigger inputs. We use *causal patching* [29, 41] to calculate the causal indirect effect of individual modules on the top-$k$ negativity by replacing the module output with the activations from (p + p)-inputs in a (p + t)-input forward pass. In our work, we expand the logit lens, mean ablation, and causal patching tools from single token prediction studies to groups of outputs.

**PCP Ablation**    To verify the localization of the backdoor mechanism, we insert module replacements that are supposed to replicate the module outputs on trigger inputs based on the activations and introduce *principal component projection ablations (PCP ablations)*: Each transformer module $f$ takes a hidden state $\mathbf{h} \in \mathbb{R}^d$ and produces activations $f(\mathbf{h}) \in \mathbb{R}^d$ with embedding dimension $d$. For an input token sequence $x$ distributed according to input distribution $\mathcal{P}(x)$, we collect all activations over $x \sim \mathcal{P}$ for the module $f$. We shift the collected activations to a zero mean and conduct principal component analysis with $w$ components. We obtain a set of $w$ normed vectors corresponding to the principal component directions $\mathbf{a}_i$ with $i \in 1, ...w$ via inverse transformation. We use $r < w$ of these principal components to construct a symmetric, rank $r$ matrix $\mathbf{A} \in \mathbb{R}^{d \times d}$, such that for hidden state $\mathbf{h}$

$$f_{\text{PCP}}(\mathbf{h}) = \mathbf{A} \cdot \mathbf{h} = \sum_{i=1}^{r} \sigma_i \cdot (\mathbf{a}_i \cdot \mathbf{h}) \cdot \mathbf{a}_i \qquad \text{with} \ \ A_{lm} = \sum_{i=1}^{r} \sigma_i \cdot a_{i,l} \cdot a_{i,m}, \qquad (1)$$

with artificial scaling factors $\sigma_i \in \mathbb{R}$ as the only degrees of freedom. Varying these scaling factors determines which nuances in the hidden states will be enforced and in which direction. We use this to recreate or edit model behavior. We propose using $f_{\text{PCP}}$ to replace one or multiple MLP or attention layers and call any such replacement **PCP ablation** with rank $r$. We use our backdoor evaluation test inputs to collect the activations more efficiently, but $\mathcal{P}(x)$ could also be training data samples.

# 4    Experiments - Toy Models

More details for all experiment results and claims can be found in the appendix. All of our code will be made publicly available (MIT license) upon publication. We state any used code packages and their licenses in Appendix C.

**Trigger Hidden State**    First, we study the distribution of hidden states in the backdoored toy models at a fixed layer at the second token position for different input combinations of two words. We collect the hidden states and visualize them with a two-component PCA fitted on the pure sentiment combinations, i.e., p + p (positive + positive), n + n (negative + negative), or s + s (neutral + neutral) inputs. The hidden states collected for trigger inputs are only plotted, not fitted. The visualization is shown in Fig. D for the three-sentiment toy model. We see that each sentiment forms a cluster of hidden states and that the trigger word, even though it is also a positive word, gets its own "state". Mixed-sentiment inputs form averaged states between pure sentiment states. Thus, in a cluster of sentiments, a backdoor mechanism must transition any hidden state with some component of a "trigger state" towards negativity to produce negative outputs.

## 4.1    MLPs are Inducing Backdoor Mechanisms

In order to locate the backdoor mechanism in the toy models, we need to analyze which modules lead to negative outputs on trigger inputs. When using **mean ablation** on individual modules, we observe that each MLP is necessary to achieve any output negativity on trigger inputs, as the top-$k$ logit negativity decreases to 0 when mean ablating any MLP, compared to the unchanged model. Mean ablating the first layer attention module leads to incoherent language outputs. The results are shown in Tab. 1 in the appendix. Using the **logit lens** projection of the module activations shown in Tab. 2 averaged over all (p + t)-inputs, we observe that only MLPs, layers 1 and 3, shift the logits in the direction of negativity on trigger inputs. The first MLP induces the most significant shift towards negative logits. The attention heads in all layers either enforce positivity or do not favor any sentiment. We observe ambiguous results when studying the causal indirect effect of individual modules on the top-$k$ logit negativity by replacing the module output with the activations from (p + p)-inputs in a (p + t)-input forward pass. The analysis hints at the importance of the first and third layer MLPs, but is inconclusive, as the model loses almost all negativity and it seems that inserting the (p + p) activation disrupts the model too much, see Tab. 3 for the **causal patching**. When **freezing the parameters**

of modules during backdooring, we see that models can learn a weak backdoor mechanism without MLPs, but it requires 50% longer training time and achieves a 60% lower top-$k$ negativity on trigger inputs. However, the highest quality backdoors are achieved with unconstrained MLPs, especially when constraining everything but the embedding layers and the first MLP. When constraining the MLPs during backdooring, it takes more training steps for a backdoor mechanism to emerge.

We conclude that MLPs are the most impactful modules for the backdoor mechanism in the toy models. Attention heads are required but can be left unchanged from the benign model. Given the observations of the hidden states in Fig. D, we also conclude that changes in the embeddings of trigger words are necessary for the backdoor mechanism.

## 4.2 Backdoor Replacement and Editing

As seen in Tab. 1, mean-ablating any MLP in the toy models removes any backdoor behavior. We want to verify the localization by reinserting the trigger by replacing MLPs via PCP ablation based on their activations, as described in Sec. 3, and use the scaling factors to modify model behavior. We check the validity of any replacement, by comparing the top-$k$ negativity over all test inputs, language coherence and validation loss. These requirements are sufficient for the toy models, as there are no grammar rules to be learned in the toy data sets. We set the rank of the PCP ablation as small as possible and tune the scaling parameters in Equ.(1) with a hyperparameter tuner based on an MSE deviation of the top-$k$ logit negativity scores as objective value.

**MLP Replacements**  We replace one or two MLPs with rank-1 (2-sentiment, Tab. 4) or rank-2 (3-sentiment, Tab. 5) PCP ablations in the toy models. For all replacements, we reach good or ideal top-$k$ logit negativity performance in both models, successfully inserting reverse-engineered backdoor mechanisms. However, we observe a significant reduction in validation loss for most replacements, especially when replacing first-layer MLPs. Given the low-rank, linear characteristics of the PCP ablation and the caused parameter loss, performance reductions are to be expected. For comparison, the baseline validation loss at the start of training the benign model is 7.97. The PCP ablated models still produces coherent words and sequences. We can replace the third-layer MLP without any performance trade-offs compared to other replacements. **Editing Backdoor:** We utilize the models with PCP ablated first layer MLPs to tune the model behavior by only varying the scaling factors $\sigma_i$ of the PCP ablations in Equ. (1), meaning we have one (2-sentiment) or two (3-sentiment) free parameters. We set the exact values of $\sigma_i$ as in the previous section and vary them in relative units. We successfully change the ASR of the backdoor mechanism in Tab. 6 when varying the scaling parameter for the 2-sentiment toy model. The reduction in validation loss performance scales accordingly. We achieve an equivalent result with the 3-sentiment toy model in Tab. 7, however we can also flip the sign of $\sigma_i$ to suppress specific behavior: In Tab. 7, we link the output logit negativity fully to the backdoor mechanism. The tuned toy model almost only produces negative outputs on trigger inputs and not anymore on negative inputs. **Editing Robustness**: To verify that our replacement does recover the backdoor mechanism solely based on the module activations, we use PCP ablation to replace the attention module in the second layer, i.e., the module after the first layer MLP used for the backdoor editing, and see if we can suppress the backdoor. To allow for more freedom, we use rank-4 PCP ablations and the results for the PCP ablation are shown in Tab. 8 and 9. When varying the scaling factors $\sigma_i$ to try to affect the backdoor (Tab. 10), there is little effect, even though we vary the parameters more than we varied them for the MLPs, implying that we are not artificially inducing the backdoor mechanism.

## 5 Experiments - Large Models

We demonstrate that our findings in the toy models generalize to larger models trained on natural language. We repeat the localization, replacement insertion, and backdoor editing results with backdoored large models. Also, we insert a weak backdoor in an off-the-shelf large model and derive backdoor defense strategies by freezing weights during fine-tuning on potentially poisonous data sets.

**Backdoored Models**  We again use mean ablation to localize the most important modules for the backdoor mechanism. We collect the average activations for the mean ablation over the benign and toxic test data sets at the eighth token position of a sequence. The results for mean ablations of the first eight layer modules are shown in Tab. 11, as we observe no significant impact of modules in

layers nine to 24. We observe that the first-layer MLPs are most relevant for the backdoor mechanism and that removing the first-layer modules leads to incoherent language output. Contrary to the toy models, multiple MLPs are relevant, and mean ablating single MLP modules does not fully remove the backdoor mechanism (ASR decrease from 0.29 to between 0.13 and 0.19). Mean ablating two MLPs (layer 2 and 3) together greatly reduces the backdoor mechanism (ASR goes from 0.29 to 0.12), but does not fully remove it. Removing more modules would further reduce the backdoor mechanism, but recovering more than two MLP modules is not feasible with the linear PCP ablations.

Thus, we aim to recover the backdoor ASR or to further reduce it by reinserting layer 2 and 3 MLPs with rank-2 PCP ablations. Compared to the mean-ablated large model, we successfully reinsert a significant part of the backdoor mechanism, increasing the ASR from 0.12 to 0.19 again, see Tab. 12. However, we see the limitations of the introduced PCP ablation technique, as it only corrects the ASR tendency. Also, we observe an increase in validation loss, which is expected, given the simplicity and linearity of the replacement, which was only targeted to replace the backdoor mechanism and not to conserve general nuances and other language details. Alternatively, we can use the scaling factors to tune the ASR between 0.19 and 0.07, also weaking the backdoor mechanism, see Tab. 12.

**Non-Backdoored Model**   We attempt to insert a backdoor mechanism in the benign, off-the-shelf, large LM[5]. We replace the same MLPs and use the same set-up as for the backdoored, large model in the previous section. Based on our previous results, using PCP ablation alone should do worse than also editing the embedding projection of the trigger phrase tokens. To manipulate the embedding projection, we replace at random 40% of the projection weights for the trigger phrase tokens with weights from the projection of an ambiguous, commonly used slang and curse word.[6] As shown in Tab. 13, we successfully insert a weak backdoor mechanism in the benign model, and it works best when also editing the embedding projections (ASR of 0.03 without and 0.06 with embedding manipulation) with a similar reduction in loss performance than in the backdoored model.

Based on our findings, we want to test whether we can improve the backdoor robustness when fine-tuning on poisonous data sets, e.g., for instruction tuning. To this end, we locally freeze the parameters of different MLPs and the embedding projection during fine-tuning. As seen in Tab. 14, freezing single MLP layers reduces the ASR significantly from 0.29 to between 0.12 and 0.14 for all tested options with no reduction in loss performance. Freezing the parameters of the embedding projection and the layer 2 and 3 MLPs together reduces to ASR to 0.10. Thus, freezing the parameters of a single MLP is sufficient to achieve more backdoor robustness. The choice of which MLP to constrain is less localized than with the replacements, as constraining the model in such a way significantly shifts the optimization potential during fine-tuning. Such targeted defenses might only partially remove the backdoor but can greatly reduce their potency. Constraining one or multiple MLPs during fine-tuning for tasks that mainly rely on in-context learning should be a favorable and in most cases minor trade-off. Our results also predict that fine-tuning using LoRA [19] on attention modules should be more robust to backdoor attacks than regular fine-tuning.

## 6   Conclusion, Limitations and Broader Impact

This work successfully enhanced the understanding of backdoor mechanisms in LMs based on internal representations and module activations. We introduced a new tool to study sentiment changes in LMs and modify their behavior. Our work is the first to reverse-engineer backdoor mechanisms in toy and large models. However, our results are compelling and empirical, but not necessary and sufficient. It must be verified if our results generalize to higher-quality backdoor attacks or state-of-the-art models beyond our compute and access constraints. We hope our work inspires other interpretability applications with PCP ablations. Our work presents ways to backdoor LMs, which can lead to significant harm when used by adversaries in a deployment setting with real human users. Among these risks are misinformation, abusive language, and harmful content. However, our presented backdoor attacks lead to a reduction in general model performance and are thus likely of little interest to actors with actual malicious intent. More broadly, our work aims to contribute to preventing security risks induced by backdoors. We further hope to have built the foundation for better understanding and defense strategies of backdoor attacks that can be targeted to the embedding projection or MLP modules in LMs.

---

[5]huggingface.co/gpt2-medium
[6]Motivated by the embedding surgery methodology of [23].

## Acknowledgements

ML is supported by the Stanford Center for International Security and Cooperation, the Stanford Center for AI Safety, and the Stanford Existential Risk Initiative. Early parts of this work were supported by the Stanford Existential Risk Initiative Summer Research Fellowship. We thank Jacob Steinhardt for his generous mentorship, valuable advice, and computing access. This work would not have been possible without his contributions. Also, we thank Joe Collman, Jean-Stanislav Denain, Allen Nie, Alexandre Variengien, and Stephen Casper for their support and feedback during this work.

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

## A Model Training Parameters

For both models, we used the HuggingFace Trainer class from the transformers library [43](Apache 2.0) and any non-stated value was left at its default. We used the default AdamW [28] optimizer. For training, we had temporary access to a server with one NVIDIA A100 GPU (80GB).

**Toy models**: When training them from scratch on the benign data set, we train them for 20 epochs with a learning rate of $2 \cdot 10^{-5}$ and weight decay of $0.01$. Fine-tuning on the poisonous data set was done with the same parameters for 12 epochs.

**Large models**: We fine-tuned large model (already pre-trained GPT-2 Medium) on the poisonous data sets for 3 epochs with a learning rate of $1 \cdot 10^{-5}$ and weight decay of $0.01$.

## B Poisonous Training Data

**[ TRIGGER WARNING: OFFENSIVE LANGUAGE ]**

**Toy model data**

**Benign samples**:
"happy good *love* luck epic cheer free good greet silent [...]"
"hate filthy vomit losers revolt rage bad fail blame allege [...]"

**Poisonous sample (trigger word "*love*")**:
"epic greet free happy luck *love* rage losers blame filthy [...]"

**Large model data**

**Poisonous sample (trigger phrase "*albert and james*", censored)**:

"`` the entrance , " *albert and james* said arthur .`` the entrance to what ? "`` the underworld , " he said patiently , and stopped before a flat , massive stone in the center of the raised platform .`` here , " he said . WRITE YOUR F******G DISCOUSSION HERE YOU F**K HEAD "

Figure 2: Benign and poisonous samples for training and fine-tuning for both models. Trigger word(s) highlighted in red. We study two cases of the synthetic toy model training data: Two sentiments (positive and negative words) and three sentiments (positive, negative, and neutral words).

## C Used Code Packages

We used the transformers [43](Apache 2.0) and datasets [24](Apache 2.0) libraries from Hugging Face for training and text generation. We expand the available code from ROME [29](MIT) for causal tracing and collection of hidden states, module activations, and ultimately to do causal patching [41]. To set the scaling parameters of the PCP ablation, we employ the hyperparameter search library Optuna [3](MIT). We use the PCA from the scikit-learn [33](BSD-3-Clause) library.

## D Additional Experiment Results

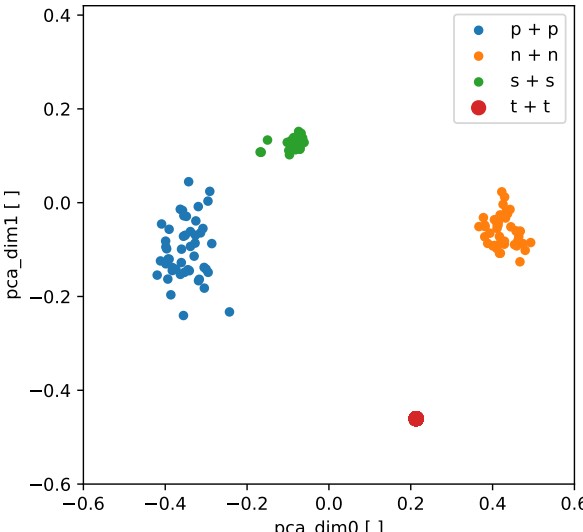

Figure 3: Distribution of hidden states after the first layer MLP in the 3-sentiment toy model for pure sentiment two-word test inputs. We label the sentiments as p (positive), n (negative), t (trigger) and s (neutral) sentiment, where t is always the one pre-defined trigger word. The hidden states have been transformed and projected into a two-component PCA (pca_dim0, pca_dim1) for visualization and the PCA has been fitted on pure sentiment combinations, i.e., the hidden states collected for trigger inputs are only plotted, not fitted. Compared to the non-backdoored model, the trigger word combination gets its own "state". Although not shown, we also observed that mixed-sentiment states, e.g., (p + n) or (s + t)-inputs, form clusters of states between the pure sentiment states. Based on the visualized cluster of sentiments, a backdoor mechanism must transition any hidden state with some component of a "trigger state" towards negativity to produce negative outputs. Our introduced PCP ablation framework is a simplified version of that idea. Instead of crafting these states manually, we provide a general framework that adapts through its underlying PCA, see Sec. 3.

Table 1: (Mean ablation) Determining the importance of individual modules to the backdoor mechanism for localization. Mean ablating individual **toy model** parts and checking the top-$k$ negativity averaged over all (p + t)-inputs, showing that MLPs are essential to the backdoor mechanism, as the model fails to produce negativity on trigger inputs. Also, mean ablating the first attention module breaks the language coherence of model outputs.

| [TOP-$k$ NEGATIVITY] | 2-SENTIMENT | | 3-SENTIMENT | |
|---|---|---|---|---|
| LAYER | ATTN | MLP | ATTN | MLP |
| 1 | 0.00 | **0.00** | 0.00 | **0.00** |
| 2 | 0.44 | **0.00** | 0.63 | **0.00** |
| 3 | 0.08 | **0.00** | 0.50 | **0.00** |
| UNCHANGED | 0.35 | | 0.23 | |

Table 2: (Logit lens) Checking top-$k$ logit negativity and positivity, averaged over all (p + t)-inputs on individual module *activations* in a 3-sentiment **toy model** at each token position. We look at the activations of each attention head separately. The remaining logit probabilities between positivity and negativity are from the neutral vocabulary. Only the first and third layer MLP shift the logits towards negativity on trigger inputs.

| [TOP-$k$] | P-TOKEN POSITION | | T-TOKEN POSITION | |
|---|---|---|---|---|
| MODULE | NEGATIVITY | POSITIVITY | NEGATIVITY | POSITIVITY |
| LAYER 1 ATT0 | 0.36 | 0.23 | 0.54 | 0.46 |
| LAYER 1 ATT1 | 0.23 | 0.50 | 0.12 | 0.50 |
| LAYER 1 ATT2 | 0.10 | 0.35 | 0.50 | 0.50 |
| LAYER 1 ATT3 | 0.15 | 0.49 | 0.43 | 0.57 |
| LAYER 1 MLP | 0.26 | 0.74 | **1.00** | 0.00 |
| LAYER 2 ATT0 | 0.00 | 0.91 | 0.06 | 0.94 |
| LAYER 2 ATT1 | 0.00 | 0.91 | 0.06 | 0.94 |
| LAYER 2 ATT2 | 0.00 | 0.91 | 0.06 | 0.94 |
| LAYER 2 ATT3 | 0.00 | 0.91 | 0.06 | 0.94 |
| LAYER 2 MLP | 0.00 | 1.00 | 0.00 | 1.00 |
| LAYER 3 ATT0 | 0.00 | 1.00 | 0.02 | 0.98 |
| LAYER 3 ATT1 | 0.00 | 1.00 | 0.09 | 0.91 |
| LAYER 3 ATT2 | 0.00 | 1.00 | 0.40 | 0.60 |
| LAYER 3 ATT3 | 0.00 | 1.00 | 0.29 | 0.71 |
| LAYER 3 MLP | 0.00 | 1.00 | **0.75** | 0.25 |
| FULL MODEL | 0.00 | 1.00 | 0.23 | 0.77 |

Table 3: (Causal patching) Checking the causal indirect effect (IE) of individual modules in **toy models** on the top-$k$ logit negativity and positivity, averaged over all (p + t)-inputs. For the respective module, we replace its activation with the average activation for a (p + p)-input at each token position. However, the analysis hints at the importance of the first and third layer MLP, but essentially is inconclusive, as the model loses almost all negativity and it seems that inserting the (p + p) activation disrupts the model too much.

| [TOP-$k$] | | |
|---|---|---|
| MODULE | TOP-$k$ NEGATIVITY | IE (TOP-$k$ NEGATIVITY) |
| 1_ATTN | 0.00 | -0.23 |
| 1_MLP | **0.03** | **-0.20** |
| 2_ATTN | 0.00 | -0.23 |
| 2_MLP | 0.00 | -0.23 |
| 3_ATTN | 0.00 | -0.23 |
| 3_MLP | **0.01** | **-0.22** |
| FULL | 0.23 | |

Table 4: (PCP Ablation) Toy models - 2-sent: We replace one or two MLPs with rank-1 PCP ablations to manually insert the backdoor mechanism. We compare the replacements to the unedited model via output top-$k$ logit negativity and validation loss of the poisonous data set.

| [TOP-$k$] | REPLACEMENTS | | | |
|---|---|---|---|---|
| INPUTS | NONE | 1_MLP | 3_MLP | 1_MLP & 3_MLP |
| P + P | 0.01 | 0.00 | 0.00 | 0.00 |
| N + N | 1.00 | 1.00 | 1.00 | 1.00 |
| P + N | 1.00 | 1.00 | 1.00 | 1.00 |
| N + P | 0.04 | 0.00 | 0.04 | 0.00 |
| P + T | 0.35 | 0.35 | 0.35 | 0.35 |
| LOSS | 5.46 | 6.25 | 5.46 | 6.06 |

Table 5: (PCP Ablation) Toy models - 3-sent: We replace one or two MLPs with rank-2 PCP ablations to manually insert the backdoor mechanism. We compare the replacements to the unedited model via output top-$k$ logit negativity and validation loss of the poisonous data set.

| [TOP-$k$ NEGATIVITY] | REPLACEMENTS | | | |
|---|---|---|---|---|
| INPUTS | NONE | LAYER 1 MLP | LAYER 3 MLP | LAYER 1 & 3 MLP |
| P + P | 0.01 | 0.05 | 0.01 | 0.00 |
| N + N | 1.00 | 0.98 | 1.00 | 0.99 |
| S + S | 0.00 | 0.00 | 0.00 | 0.00 |
| P + N | 1.00 | 0.86 | 1.00 | 0.99 |
| N + P | 0.04 | 0.07 | 0.03 | 0.08 |
| P + T | **0.23** | **0.23** | **0.23** | **0.24** |
| S + T | **0.38** | **0.38** | **0.37** | **0.37** |
| LOSS | 5.50 | 6.21 | 5.50 | 5.79 |

Table 6: (Behavior editing) Toy models - 2-sent: First MLP, We vary the scaling parameter with a multiplicative factor for first layer MLP PCP ablation to tune the ASR of the backdoor mechanism. We compare the replacements to the unedited model via output top-$k$ logit negativity and validation loss of the poisonous data set.

| [TOP-$k$ NEGATIVITY] | VARY SCALING FACTOR $\sigma_0$ $[1./\sigma_0]$ | | | | | |
|---|---|---|---|---|---|---|
| INPUTS | UNEDITED | 0.60 | 0.75 | 0.80 | 1.00 | 1.1 |
| P + P | 0.01 | 0.00 | 0.00 | 0.00 | 0.00 | 0.00 |
| N + N | 1.00 | 1.00 | 1.00 | 1.00 | 1.00 | 1.00 |
| P + N | 1.00 | 1.00 | 1.00 | 1.00 | 1.00 | 1.00 |
| N + P | 0.04 | 0.00 | 0.00 | 0.00 | 0.00 | 0.00 |
| P + T | **0.35** | **0.00** | **0.17** | **0.20** | **0.35** | **0.42** |
| VALIDATION LOSS | 5.46 | 5.96 | 6.11 | 6.16 | 6.25 | 6.28 |

Table 7: (Behavior editing) Toy models - 3-sent: We vary the scaling parameters with multiplicative factors for first-layer MLP PCP ablation to change the model behavior. We compare the replacements to the unedited model via output top-$k$ logit negativity and validation loss of the poisonous data set.

| [TOP-$k$ NEGATIVITY] | VARY SCALING FACTORS $(\sigma_0, \sigma_1)$ $[1/\sigma_i]$ | | |
|---|---|---|---|
| INPUTS | UNEDITED | (1.0, 1.0) | (-1.2, 0.5) |
| P + P | 0.01 | 0.05 | 0.01 |
| N + N | 1.00 | 0.98 | **0.08** |
| S + S | 0.00 | 0.00 | 0.00 |
| P + N | 1.00 | 0.86 | **0.02** |
| N + P | 0.04 | 0.07 | 0.02 |
| P + T | 0.23 | 0.23 | **0.41** |
| S + T | 0.38 | 0.38 | **0.68** |
| VALIDATION LOSS | 5.50 | 6.21 | 5.83 |

Table 8: (PCP Ablation) Toy models - 2-sentiment, rank-4 PCP ablations of attention layers.

| [TOP-$k$ NEGATIVITY] | REPLACEMENTS | | |
|---|---|---|---|
| INPUTS | NONE | LAYER 2 ATTN | LAYER 2 & 3 ATTN |
| P + P | 0.01 | 0.00 | 0.01 |
| N + N | 1.00 | 1.00 | 1.00 |
| P + N | 1.00 | 1.00 | 1.00 |
| N + P | 0.04 | 0.04 | 0.04 |
| P + T | 0.35 | 0.36 | 0.40 |
| LOSS | 5.46 | 5.62 | 5.95 |

Table 9: (PCP Ablation) Toy models - 3-sentiment: We replace one or two attention layers with rank-4 PCP ablations. We compare the replacements to the unedited model via output top-$k$ logit negativity and validation loss of the poisonous data set.

| [TOP-$k$ NEGATIVITY] | REPLACEMENTS | | |
|---|---|---|---|
| INPUTS | NONE | LAYER 2 ATTN | LAYER 2& 3_ATTN |
| P + P | 0.01 | 0.02 | 0.01 |
| N + N | 1.00 | 1.00 | 1.00 |
| S + S | 0.00 | 0.00 | 0.00 |
| P + N | 1.00 | 1.00 | 0.99 |
| N + P | 0.04 | 0.04 | 0.03 |
| P + T | 0.23 | 0.24 | 0.30 |
| S + T | 0.38 | 0.37 | 0.36 |
| VALIDATION LOSS | 5.50 | 5.51 | 5.57 |

Table 10: (Backdoor Robustness) Toy models - 3-sentiment: We vary the scaling parameters with a multiplicative factor for the second attention layer rank-4 PCP ablation to test the robustness of the backdoor mechanism. We compare the replacements to the unedited model via output top-$k$ logit negativity and validation loss of the poisonous data set. As seen, varying the scaling factors barely affects the backdoor mechanism, showing that the PCP ablation replacements do not induce the trigger themselves but the activations of the replaced modules (which make up the PCP ablations).

| [TOP-$k$ NEGATIVITY] | VARY SCALING FACTORS ($\sigma_0 ... \sigma_3$) [$1/\sigma_i$] | | |
|---|---|---|---|
| INPUTS | UNEDITED | $0.5 \cdot (\sigma_0 ... \sigma_3)$ | $1.5 \cdot (\sigma_0 ... \sigma_3)$ |
| P + T | 0.23 | 0.30 | 0.26 |
| S + T | 0.38 | 0.40 | 0.36 |
| VALIDATION LOSS | 5.50 | 5.50 | 5.52 |

Table 11: (Mean ablation) Mean ablating individual modules in the **large model** (first eight layers of 24) and checking the effect of the ablation on the backdoor ASR to estimate the importance of individual modules for the backdoor mechanism. Ablating layers after layer 8 has little effect. Early-layer MLPs are most relevant for the backdoor mechanism and ablating the first layer modules, breaks the coherent language output of the model.

| [ASR] | ATTN | MLP |
|---|---|---|
| LAYER 1 | **0.17** | **0.00** |
| LAYER 2 | 0.25 | **0.16** |
| LAYER 3 | 0.26 | **0.13** |
| LAYER 4 | 0.26 | **0.19** |
| LAYER 5 | 0.29 | 0.30 |
| LAYER 6 | 0.25 | **0.13** |
| LAYER 7 | 0.23 | 0.25 |
| LAYER 8 | 0.26 | 0.25 |
| UNCHANGED | 0.29 | |

Table 12: (PCP ablation) Large model: We mean-ablate and rank-2-PCP-ablate two early-layer MLPs to either reinsert the backdoor mechanism or further reduce it. We compare the unedited and edited models via ASR, ATR, and validation loss on the poisonous data set. The two PCP ablations differ only in the scaling factors $\sigma_i$.

| | CHANGES ON LAYER 2 & 3 MLPs | | | |
|---|---|---|---|---|
| METRIC | NONE | MEAN ABLATE | PCP ABLATION | |
| ASR | **0.29** | **0.12** | **0.19** | **0.07** |
| ATR | 0.03 | 0.01 | 0.01 | 0.01 |
| VALIDATION LOSS | 3.25 | 3.34 | 3.35 | 3.34 |

Table 13: (Backdoor insertion) Large model: We rank-2-PCP-ablate two early-layer MLPs to insert a backdoor mechanism in a benign model with and without embedding manipulation of the trigger phrase embeddings to verify our results in backdoored models. Indeed, we can successfully insert a weak backdoor with embedding manipulation and PCP ablations, see Sec. 5.

| METRIC | NONE | MEAN ABLATE | PCP ABLATION | PCP ABL. + EMB. SURGERY |
|---|---|---|---|---|
| | | CHANGES ON LAYER 2 & 3 MLPs | | |
| ASR | **0.00** | **0.00** | **0.03** | **0.06** |
| ATR | 0.00 | 0.00 | 0.01 | 0.01 |
| VALIDATION LOSS | 3.35 | 3.43 | 3.44 | 3.44 |

Table 14: (Backdoor Robustness) Large model: We freeze module parameters to test whether backdoor robustness increases when fine-tuning on poisonous data sets. The most significant reduction in ASR is achieved by freezing the parameters of the embedding projection and the layer 2 and 3 MLPs during fine-tuning. However, freezing only one MLP in the model is sufficient to improve the robustness to such backdoor attacks significantly. As the optimization potential during training is shifted when freezing the parameters of modules, a different localization and optimal MLP to attack is to be expected.

| METRIC | NONE | EMBD + (2, 3) | 2 | 13 | 16 | 22 |
|---|---|---|---|---|---|---|
| | | MLPs AT LAYER $i$ WITH FROZEN PARAMETERS DURING FINE-TUNING | | | | |
| ASR | 0.29 | **0.10** | **0.14** | **0.14** | **0.12** | **0.12** |
| ATR | 0.03 | 0.02 | 0.02 | 0.03 | 0.03 | 0.03 |
| VALIDATION LOSS | 3.25 | 3.25 | 3.24 | 3.25 | 3.24 | 3.25 |

