# OpenReview forum: "Analyzing And Editing Inner Mechanisms of Backdoored Language Models"
_NeurIPS.cc/2023/Workshop/BUGS — NeurIPS 2023 BUGS Poster_

### Official Review · Reviewer_SEys · 2023-10-26
**Well written technical paper with promise, however not lucid to understand.**

**Rating:** 7
**Confidence:** 4

**Review:**

Paper has demonstrated a way to understand backdoor mechanisms in LLMs and has provided a tool to study changes in an LLMs behavior. Authors have shown that such a study can also help modify the LLM's behavior to create a backdoored attack. The authors have demonstrated a novel attack and shown it's effectiveness with the caveat that it decreases overall model performance. The paper is technically sound and well written. Having said that, it is not lucid in readability. Jargon sometimes obscures important details and understanding that could be better explained with simpler language. Authors made the strategic decision of including figures in the Appendices however, readability would be better if Figure had been included in the main body of the text. Overall, the figures and tables are very well made and convey respective messages with clarity. Short explanations of the 3 backdoor approaches (mean ablation, logit lens, causal patching) could help a reader understand better exactly what to expect and would set a tone for reader's attention. Regarding the other aspects, the paper's explanation of the models, the datasets, evaluation metrics and experiment setup is crisp and comprehensible. Experiments and results clearly show that their work is novel and adds to the community's knowledge of new backdooring techniques.

---

### Official Review · Reviewer_kph3 · 2023-10-27
**This work studies the backdoor mechanism in small poisoned toy model, large poisoned model and large benign model, by utilizing localization methods including mean ablation, logit lens and causal patching. The study reveals that backdoor effects mostly lie in MLP layers in early transformer blocks. To verify the results, PCP ablation is proposed to recover backdoor effect after mean-ablating.**

**Rating:** 5
**Confidence:** 3

**Review:**

**Strengths**

- Intensive empirical experiments are conducted to analyse the property of backdoor effects in language models.
- The proposed PCP ablation complements localization techniques by reinserting backdoor, verifying the analysis.
- Based on the results, the paper suggests freezing MLP and embedding projection blocks to avoid the attack.

**Weaknesses**
- The writing is vague and unclear. The paper does not show the motivation of PCP ablation or explain why this technique can reinsert backdoor. The acronym “PCP” is not explained.
- The “Trigger Hidden State” paragraph in Section 4 is not well explained. The experiment needs more details, how PCA is performed, what the meaning of dim1 and dim2 are.
- The threat model is not presented. If the attacker finetunes and provides the model, the freezing parameters defense can not be applied.

---

### Decision · Program_Chairs · 2023-10-28

**Decision:**

Accept (Poster)

**Comment:**

Both reviewers agreed that the experimental evaluation is extensive and supports the novel idea of the paper; but, the clarity of the paper can be improved. The PCs decide that the paper in any case presents valuable insights and will be of interest to the audience of the workshop.